# Machine Learning-Based Prediction of Selected Parameters of Commercial Biomass Pellets Using Line Scan Near Infrared-Hyperspectral Image

Lakkana Pitak [1], Kittipong Laloon [1,2], Seree Wongpichet [1], Panmanas Sirisomboon [3] and Jetsada Posom [1,4,*]

1 Department of Agricultural Engineering, Faculty of Engineering, Khon Kaen University, Khon Kaen 40002, Thailand; lakkana_p@kkumail.com (L.P.); kittila@kku.ac.th (K.L.); serwon123@gmail.com (S.W.)
2 Agricultural Machinery and Postharvest Technology Center, Khon Kaen University, Khon Kaen 40002, Thailand
3 Department of Agricultural Engineering, School of Engineering, King Mongkut's Institute of Technology Ladkrabang, Bangkok 10520, Thailand; panmanas.si@kmitl.ac.th
4 Applied Engineering for Important Crops of the North East Research Group, Department of Agricultural Engineering, Faculty of Engineering, Khon Kaen University, Khon Kaen 40002, Thailand
* Correspondence: jetspo@kku.ac.th; Tel.: +66-43-009700

**Abstract:** Biomass pellets are required as a source of energy because of their abundant and high energy. The rapid measurement of pellets is used to control the biomass quality during the production process. The objective of this work was to use near infrared (NIR) hyperspectral images for predicting the properties, i.e., fuel ratio (FR), volatile matter (VM), fixed carbon (FC), and ash content (A), of commercial biomass pellets. Models were developed using either full spectra or different spatial wavelengths, i.e., interval successive projections algorithm (iSPA) and interval genetic algorithm (iGA), wavelengths and different spectral preprocessing techniques. Their performances were then compared. The optimal model for predicting FR could be created with second derivative (D2) spectra with iSPA-100 wavelengths, while VM, FC, and A could be predicted using standard normal variate (SNV) spectra with iSPA-100 wavelengths. The models for predicting FR, VM, FC, and A provided $R^2$ values of 0.75, 0.81, 0.82, and 0.87, respectively. Finally, the prediction of the biomass pellets' properties under color distribution mapping was able to track pellet quality to control and monitor quality during the operation of the thermal conversion process and can be intuitively used for applications with screening.

**Keywords:** fuel ratio; proximate data; NIR hyperspectral imaging; wavelength selection; biomass pellet; in-line measurement

## 1. Introduction

Recently, the world requirement for renewable energy has increased [1]. In 2020, the use of renewable energy increased to 3.9% from the previous year [2]. Biomass is considered a renewable energy source because it is recycled and reused [3]. The use of biomass can be environmentally friendly because biomass energy can make a carbon balance, as burned hydrocarbons can release carbon dioxide ($CO_2$) into the air, and $CO_2$ can rotate again for plant photosynthesis [4]. Therefore, biomass feedstock can be produced and reused again and again. Biomass is obtained from dedicated energy crops, agricultural crop residues, forestry residues, wood processing residues, municipal waste, and wet waste [5]. This means that biomass materials differ in their physical properties, energy content, and chemical content. Most biomass energy is used as combustion fuel. It is combusted to generate electricity as a source of thermal energy for an industrial purpose [6]. Therefore, the quality of the biomass needs to be known for the quality control of the thermal conversion process, as well as for commercial and pellet production.

At present, the biomass patterns used for combustion include ground biomass, biomass chips, and biomass pellets. Ground and wood chip biomass are easy to prepare, but they are difficult to store and transport, and their moisture content (MC) is very variable. However, biomass pellets provide many advantages such as having a consistent size, lower moisture content, higher calorific density, and convenience of transportation [7].

To confirm the stable development of biomass pellets, the quality of commercial pellets must be maintained in the production line, which depends on the raw materials used. Therefore, a rapid and accurate method for determining pellet quality should be used to select the pellet products that provide a good combustion performance. This can achieve the efficient utilization of biomass. For commercial purposes, the quality of biomass pellets must follow international standards, i.e., European standards for pellet quality [8]. The important parameters of biomass pellets in the production process are density, strength, and moisture content (MC) [9].

For the utilization of biomass, the parameters of proximate analysis and calorific value are essential [10]. Biomass pellets with a high level of volatile matter (VM) can be flammable and could be prepared for pyrolysis and gasification, while biomass pellets with a high fixed carbon (FC) give high combustion heating [7] and could be used for direct combustion and slow pyrolysis. The ratio of FC to VM is called the "fuel ratio (FR)" and is an important parameter used to manage the ratio of syngas and combustion air [11]. From a study of the burning rate of coal, it was found that coal can be easily burned if the fuel ratio is lower than 1.5, but it is difficult to ignite if the FR is higher than 1.5. Therefore, knowledge of the FR can improve the efficiency of thermal energy combustion [12]. Ash (A) is the residue left after combustion and is a factor used for designing combustion furnaces [10]. A low A content in biomass material can reduce the cost of disposal [7].

However, the traditional measurement methods mentioned above, though accurate are time-consuming because of their complexity and lead to labor and chemical costs. This means that the biomass quality cannot be checked before the commercial and operation of the thermal conversion process (such as combustion, pyrolysis, and gasification). Additionally, pellet quality cannot be followed up because the measurement method cannot be checked in real time. Therefore, a rapid measurement that can be applied for in-line measurement is necessary. The parameters of proximate data should always be checked during the operation of energy conversion systems. Therefore, in-line measurement is essential because it allows the operators to know the qualities of a pellet on a real-time basis. In the case of the user, knowing the quality of biomass in real time during combustion can improve the management of possible effectiveness [7] and biomass loading.

Nowadays, near infrared (NIR) spectroscopy is used in the quality the measurement of biomass, such as for the MC and calorific value (CV) of the biomass pellets of rice stalks, rice husks, mahogany wood, tea trees, tropical wood, pine wood, rubber wood, and mixtures of wood material [13]; the pyrolysis characteristics of milled bamboo [14]; the proximate data and lignocellulose components (cellulose, semi-cellulose, and lignin) of corn stover using near infrared spectroscopy (NIRS) [15]; the MC, CV, A and carbon content (C) of *Miscanthus* and short rotation coppice Willow (SRCW) biomass [16]; and biomass pellet (wood, *Miscanthus*, and herbaceous energy grasses) quality indices (moisture, carbon, and ash contents and gross calorific value) [17]. It has been noted that NIR spectroscopy can be applied to assess the composition of analytes. This is an indirect method of measurement, and its accuracy cannot be equal to that of the laboratory method [18,19]. The application of NIR is acceptable if the error does not exceed the control limit.

An NIR hyperspectral image can be used like NIR spectroscopy, but it has the advantage of high resolution and can be represented in the form of a mapping distribution, because its precision is based on pixel resolution. Two-dimensional NIR spectroscopic imaging is applied for predicting the quality of biomass pellets, such as the CV and proximate data of biofuel pellets [13]; predicting the gaseous and particulate matter emissions of pine wood pellets [20]; and assessing the MC, specific energy, and feed rate of pelleting biomass feedstock [21]. It has thus been recommended as having the potential for applica-

tion in the biomass pelleting industry for real-time measurement to improve the efficiency of the system [21]. However, NIR hyperspectral image spectroscopy also has many factors that influence its precision and accuracy, especially the model development process, which requires improvement. In model development, wavelength and spectral pretreatments must be matched with the analyte. If the model creation process is suitable, the model can provide a high accuracy [13]. The wavelength selection is very important for improving the model's capabilities. Geoffrey et al. [22] stated that the correct wavelength can increase the temporal resolution and maintain acceptable levels of uncertainty in the predicted value. In the industrial context, optimal wavelength selection can reduce the number of wavelengths but can maintain their accuracy, which is preferable for industrial settings equipped with low-cost spectrometers [23]. Several spectra variables are selected from the full wavelength region and can be applied to develop a simple spectrometer system to detect the characteristics of the material of interest [24,25]. The main contribution of the application of NIR hyperspectral imaging could be in enabling the application of a machine learning algorithm to find the optimal wavelength selection and improve its ability based on spectral pro-processing.

The main objective of this research was to predict the FR and proximate data of commercial biomass pellets using NIR hyperspectral imaging. The sub-objectives included to compare the model performance developed from different spatial wavelengths (among full wavelength, interval successive projections algorithm (iSPA), and internal genetic algorithm (iGA) wavelengths) and spectral pretreatment methods (including raw, the first derivative (D1), second derivative (D2), and standard normal variate (SNV)), as well as to select the optimal prediction models. This research will benefit biomass pellet trading and thermal conversion processes by achieving better efficiency and an improved pelletization process at the stage of drying or storing the raw material.

## 2. Materials and Methods

### 2.1. Investigated Parameters

The main part of this study was based on estimation of selected important parameters of biomass pellets in terms of their suitability for energy production. Therefore, the investigated parameters (FR, VM, FC, and A) were determined at the stage of assessing the suitability of biomass for energy purposes and for conversion into biofuel. The standardized biomass pellet samples, with standardized parameters such as a specific density of about 1000 kg/m$^3$, mechanical durability of about 95%, and moisture content <10%, were used in experiments to determine the FR, VM, FC, and A. This uniformity facilitates the measurement of NIR hyperspectral imaging and the selected parameters.

### 2.2. Sample

A total of 140 biomass pellet samples were obtained from the wood pellet industry. Biomass pellets including filter cake (11 pellets), *Leucaena leucocephala* (9 pellets), bamboo (9 pellets), cassava rhizome (15 pellets), bagasse (14 pellets), sugarcane leaves (15 pellets), straw (15 pellets), rice husk (14 pellets), eucalyptus bark (11 pellets), Napier grass (13 pellets), and corn cob (14 pellets) were used for the experiments. The pellets used as test material were assumed to have a moisture content in the range of 6–10%. Therefore, the model developed assumed the MC as a constant that did not affect the obtained results. The biomass was pelletized using a pelletizer machine (KN-D-200, Tianjin, China), and the pellets were kept in a zipper bag and stored at room temperature 25 ± 2 °C.

### 2.3. NIR Hyperspectral Image Measurement

Figure 1 shows the pellet placed on a sample holder, made from acrylic sheet: a translucent plastic sheeting with a size of 14 × 20 cm. The sample holder was then placed on a plastic box, its bottom painted black and sized 14 × 20 × 15 cm, as a different color from the pellet sample to make it easy to indicate the region of interest (ROI). The measurement systems are shown in Figure 2, including an NIR hyperspectral camera

(Imspector N17E; Specim, Finland), a CCD camera (Xeva 992; Xenics Infrared Solutions, Leuven, Belgium), two 500 W tungsten halogen lights (Lowel Light Inc., Hauppauge, NY, USA), control software (Specim's LUMO Software Suite; Spectral Imaging Ltd., Oulu, Finland), 320 pixels (with a resolution of 30 μm per pixel) in the *x*-axis and 497 pixels in the *y*-axis. The sample was scanned using a wavelength range between 900 and 1700 nm, with a spectral resolution of 3.2 nm and a translation stage at a speed of 10 mm/s.

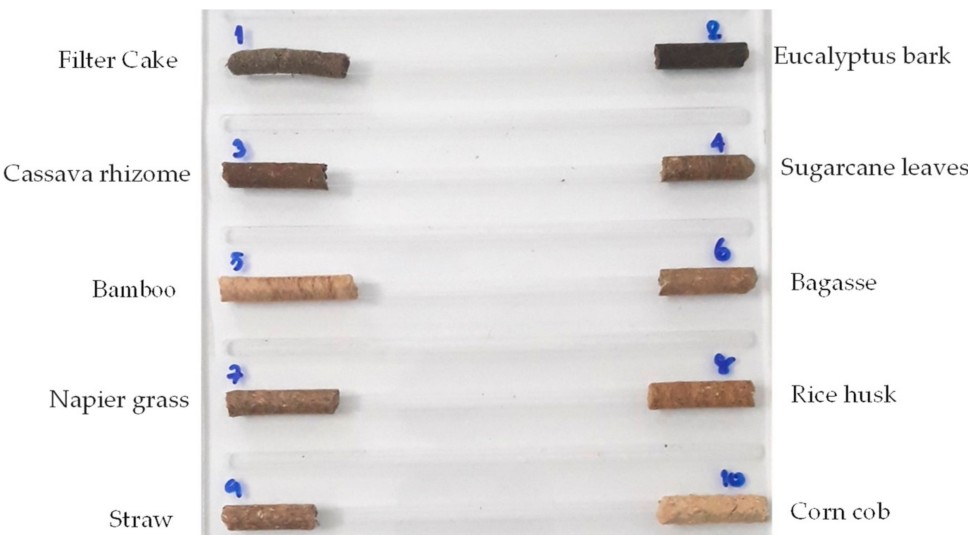

**Figure 1.** Biomass pellet samples placed on a sample holder.

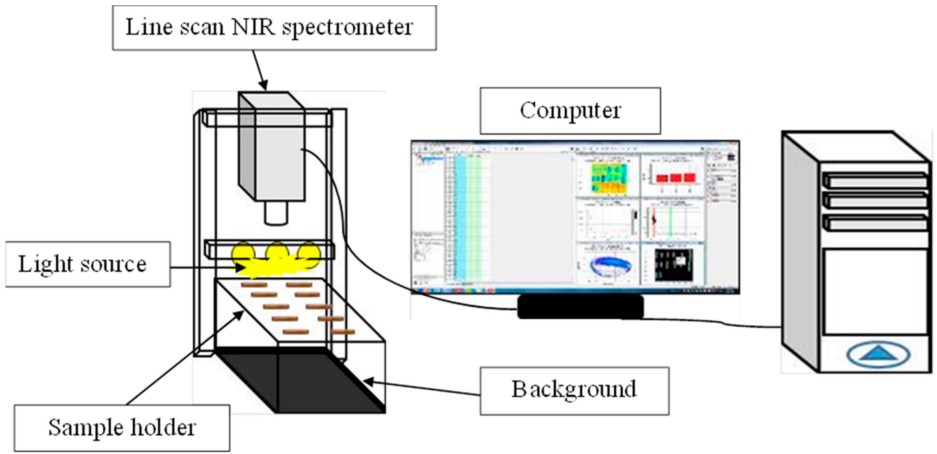

**Figure 2.** Apparatus of near infrared (NIR) hyperspectral imaging systems.

After scanning, the relative reflectance (R) of the sample was then calculated as $\frac{R_{sample} - R_{dark}}{R_{white} - R_{dark}}$ [26], where $R_{sample}$, $R_{dark}$, and $R_{white}$ are the intensity reflectance of the sample, dark reference, and white reference, respectively. R was implemented for modelling, which is the independent variable.

### 2.4. Reference Methods

After scanning, each pellet sample was ground and then used to determine the analyte including the FR, VM, FC, and A. These were determined through TGA (TG 209 F3 Tarsus, Netzsch, Germany). The ground biomass pellets of approximately 20 mg were placed in a cup of aluminum oxide ($Al_2O_3$), and the sample was heated with temperatures from 30 to 900 °C, with a heating rate of 10 K/min and an $O_2$ flow rate of 20 mL/min. After combustion, the thermogravimetric (TG) curve was presented as a line plot between mass

and temperature. The percentage of VM, FC, and ash was determined by considering the slope to slope method, the so-called slope method. The TG profile and differential thermogravimetric (DTG) profiles were obtained using Proteus 6.0.0 (Netzsch Software, Gerätebau GmbH, Nordstadt, Germany). The fuel ratio was calculated as:

$$\text{Fuel ratio} = \frac{\text{FC}}{\text{VM}} \qquad (1)$$

To check the precision of the reference method, the standard error of laboratory (SEL) of VM, FC, and A examined and were calculated as [27]:

$$\text{SEL} = \sqrt{\frac{\sum_{i=1}^{n} \left(D - \overline{D}\right)^2}{n - 1}} \qquad (2)$$

where D is the difference between duplicates, $\overline{D}$ denotes the average difference between duplicates, $\left(\frac{\sum_{i=1}^{n} D^2}{n}\right)$, and n denotes the number of samples. The SEL defines the standard deviation of the difference value between duplicates. The SEL is used to calculate the maximum coefficient of determination ($R^2_{max}$) [27], as follows:

$$R^2_{max} = \frac{SD_y^2 - SEL^2}{SD_y^2} \qquad (3)$$

where $SD_y$ is the standard deviation of the reference value in the calibration set and $R^2_{max}$ has a maximum in ($R^2$) if there is no error in spectral acquisition [27]. $R^2_{max}$ depends on the $SD_y$ and SEL. A lower $R^2_{max}$ means the sample reference method should be improved or, in other words, the reference method is not accurate [27].

### 2.5. Model Development and Validation

Figure 3 shows the flow chart of model development for the prediction of FR, VM, FC, and A using NIR hyperspectral imaging. The ROI in a pellet sample was collected by removing the background using principal component analysis (PCA). A thresholding method was used to separate the pixels of the pellet from the background. The spectra of the ROI of each pellet were averaged to one spectrum, which was used as a representative spectrum of the pellet, and this data were used for model construction [28,29]. To examine the best model, different spatial wavelength ranges and spectral pretreatments were studied. The raw spectrum and preprocessed spectra including the D1, D2, and SNV methods [30,31] were investigated. For example, the D1 and D2 were able to eliminate the baseline offset problem. The SNV could decrease multiplicative scattering effects [23,32]. The spectra pretreatment helped to increase the model accuracy. Either a full wavelength (256 wavelengths) or spatial wavelength was selected using a machine learning algorithm; SPA with 12, 25, 50, 100, and 150 wavelengths and GA with 12, 25, 50, 100, and 150 wavelengths [33] were investigated. The calibration models were developed using partial least squares (PLS) regression and validated using leave-one-out cross-validation [34].

After the best model was created, it was tested again to confirm whether this model could be used for future samples. The total samples were divided into a calibration set (75% of total samples) and prediction set (25% of total samples), and then they were validated again using the validation set. This information could be used, and NIR hyperspectral image analysis was carried out using MATLAB (R2019b, 40846673).

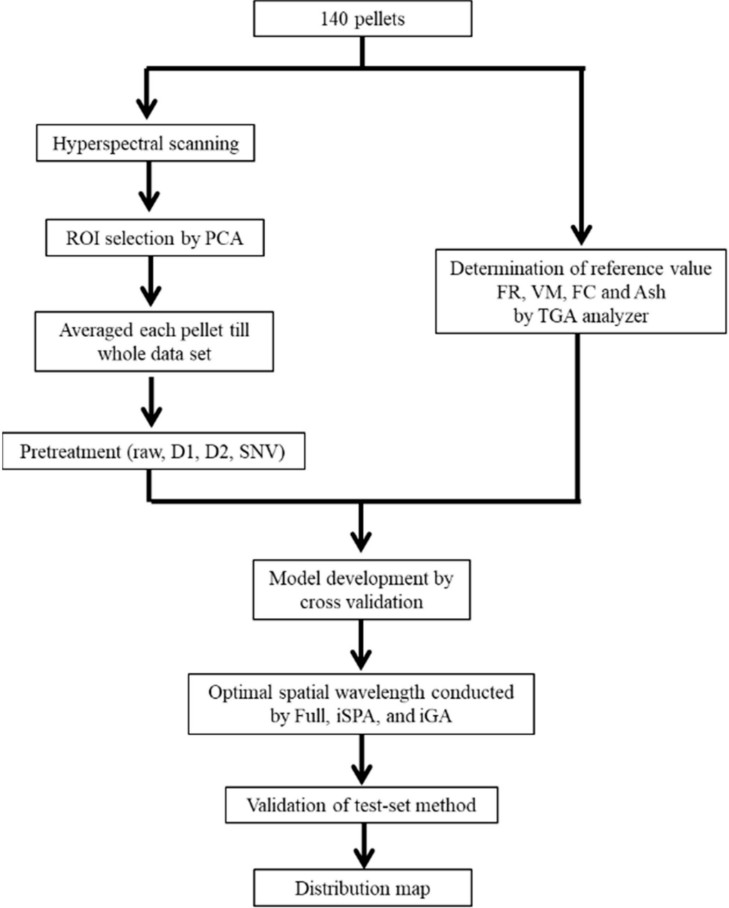

**Figure 3.** Flow diagrams for prediction of fuel ratio (FR), volatile matter (VM), fixed carbon (FC), and ash (A) by hyperspectral imaging. ROI: region of interest.

*2.6. Visualisation of FR, VM, FC, and A in the Distribution Map*

The calibration model developed from optimal conditions was applied to evaluate the analyte value of the pellet samples. Each reflection spectrum corresponding to each pixel was multiplied with the regression coefficient (B) obtained from the calibration model to calculate the predicted value, and it was represented as a 2D array of pixels. The predicted value of each pixel was calculated as $y_{pre} = x \times B$, where x is the matric of the relative reflectance of each sample. The predicted value of each pellet ($Y_{pred}$) was calculated as $Y_{pre} = \frac{\sum_1^n y_{pre}}{n}$, where n denotes the number of all pixels.

Therefore, the predicted FR, VM, FC, and A were illustrated in a 2D image levelled into different colors as a distribution map. This levelling color information can be used for visualizing the pellet quality in process and control.

After the prediction, the model's ability was then determined in the statistical term of the coefficient of determination of the prediction set ($R^2$), and the ratio of the standard error of prediction to the standard deviation of prediction (RPD) was calculated as:

$$R^2 = 1 - \frac{\sum(y_i - y_{pre})^2}{\sum(y_i - \overline{y})^2} \tag{4}$$

$$RPD = \frac{SD}{SEP} \tag{5}$$

$$\text{SEC or SEP} = \sqrt{\frac{\sum_1^n \left( y_i - y_{pre} - \frac{\sum_1^n \left( y_i - y_{pre} \right)}{n} \right)^2}{n-1}} \tag{6}$$

where $y_i$, $y_{pre}$, $\overline{y}$, SD, SEC, and SEP denote the measured value, predicted value, average measured value, standard deviation of the measured value, standard error of calibration, and standard error of prediction, respectively. The predicted results were excellent if $R^2 > 0.90$ and RPD > 3, it was a good prediction if $0.81 < R^2 < 0.90$ and $2.5 < RPD < 3$, it only permitted approximate predictions if $0.66 < R^2 < 0.80$ and $2.0 < RPD < 2.5$, and it poorly predicted if $R^2 < 0.66$ and RPD < 2 [35–37].

## 3. Results and Discussion

### 3.1. NIR Spectra

Figure 4 shows the average D2 spectra of this study of biomass pellets. For the reflectance value, negative peaks of the D2 spectra were considered as the important peaks of the absorption of the vibrated matter of interest. The derivative spectra could occur for peaks that overlapped. The different biomass types gave spectra with similar structures. However, the high negative peaks observed in the D2 relative reflectance values were found to be 940 nm, and the spectra changed with the type of pellet, which is the structure of the C-H third overtone and is associated with C-H and $CH_2$ [38]. The average D2 spectrum of rice husk showed a peak at 1100 nm, corresponding to the C-H structure and associated with the C-H aromatic [38]. The rice husk had a low reflectance value at 1250 nm, and that of the filter cake was high at 1390 nm, representing the structure of SiOH [38], which indicated that it was possible that the filter cake had more SiOH in its ash. The wavelength at 1520 nm was the vibration band of the structure of N-H (first overtone) and was associated with N-H -$CONH_2$ [38], while 1630 nm was the vibration band of the structure of C-H (first overtone) and associated with C-H=$CH_2$ [38].

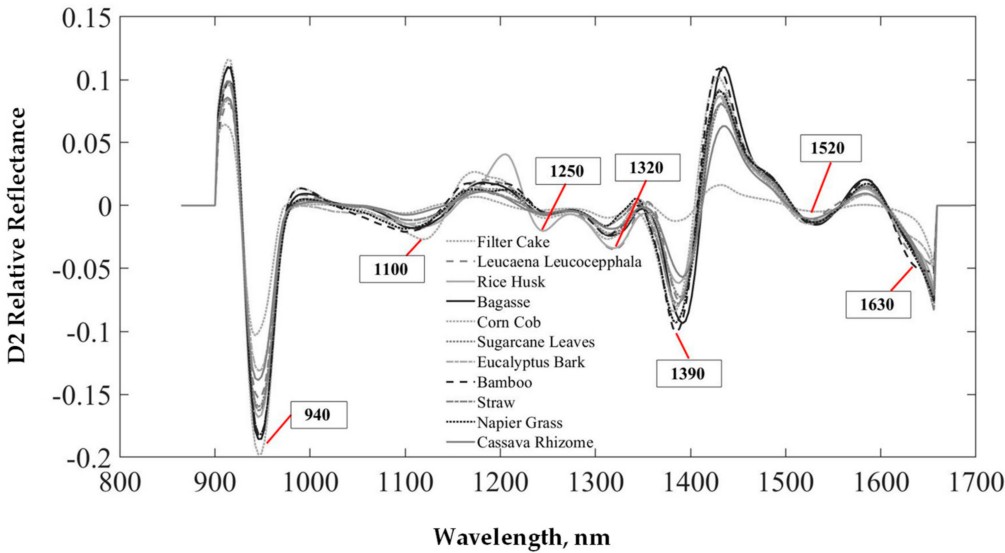

**Figure 4.** Average spectra from 11 types of the second derivative of various biomass pellets.

Similar results were also reported by Yang et al. [39] for mature bamboo (two years old) and juvenile bamboo (one month old), with many absorption bands in the wavelength region of 1100–2500 nm, including peaks at approximately 1473, 1925, 2095, 2267, and 2328 nm. Similar results were also reported by Feng et al. [16], where a large variability in reflectance values was negatively observed among different biofuel pellets for some NIR regions. The negative position of reflectance was specific at 1200, 1350, and 1450 nm [13].

### 3.2. Reference Value

Table 1 shows the statistical properties of all the biomass pellets, including FR, VM, FC, and A. FR, VM, FC, and A were in the ranges of 0.301–0.677, 39.2–66.77%, 12.83–35.23%, and −0.09–41.57%, respectively. Figure 5 shows the average FR, VM, FC, and A of the biomass pellets. The proximate data of the biomass pellets varied with different varieties. The results were similar to what was reported by Feng et al. [16]. VM was the major component of the dried biomass samples. The VM ranged from 68.72% to 89.04%, A ranged from 0.26% to 15.94%, and FC varied from 10.39% to 21.22%.

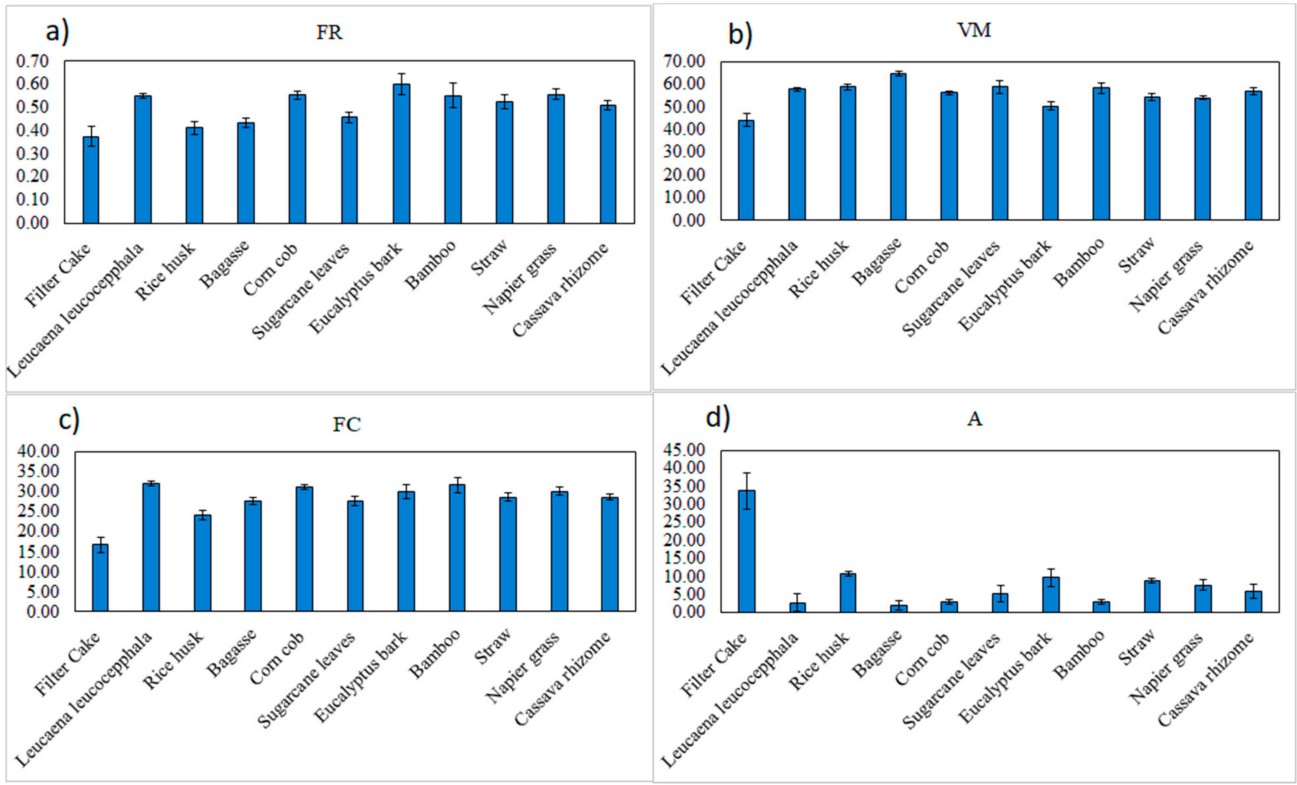

**Figure 5.** The (**a**) fuel ratio (FR, db%), (**b**) volatile matter (VM, db%), (**c**) fixed carbon (FC, db%), and (**d**) ash content (A, db%) of the 11 varieties of biomass pellets.

**Table 1.** The statistical data of fuel ratio (FR), volatile matter (VM), fixed carbon (FC), and ash (A) of all biomass types obtained by a TGA analyzer.

| Parameters | N | Max | Min | Mean | Range | SD |
|---|---|---|---|---|---|---|
| FR, db% | 140 | 0.68 | 0.30 | 0.51 | 0.38 | 0.07 |
| VM, db% | 140 | 66.77 | 39.20 | 56.71 | 27.57 | 5.17 |
| FC, db% | 140 | 35.23 | 12.83 | 28.89 | 22.40 | 4.17 |
| A, db% | 140 | 41.57 | 1.40 | 7.42 | 40.17 | 8.01 |

Table 2 shows the SEL and $R^2_{max}$ of the measured values. The SEL equaled 0.01, 0.65, 0.33, and 1.02 for FR, VM, FC, and A, respectively. The $R^2_{max}$ values of FR, VM, FC and A were 0.98, 0.99, 0.99, and 0.98, respectively. The SEL and $R^2_{max}$ are required before model development [40–42] because these parameters help to decide whether any model can be developed. If the SEL is very high, the reference method is not precise. The reference method must be re-checked; for example, it is necessary to check the scientist's skill, work instructions, chemicals, material used, and methodology [14].

**Table 2.** The standard error of laboratory (SEL) and maximum coefficient of determination ($R^2_{max}$) of fuel ratio (FR), volatile matter (VM), fixed carbon (FC), and ash (A).

| Parameter | SEL | $R^2_{max}$ |
|---|---|---|
| FR, % | 0.01 | 0.98 |
| VM, % | 0.65 | 0.99 |
| FC, % | 0.33 | 0.99 |
| Ash, % | 1.02 | 0.98 |

### 3.3. Result of Model Development

Table 3 shows the effective performance of the model in the prediction of FR, VM, FC, and A based on different spatial wavelengths, numbers of wavelengths, and spectral preprocessing techniques, where the models were validated using full cross-validation.

**Table 3.** The result of optimal wavelength selection conducted with full wavelength, interval successive projections algorithm (iSPA) wavelength, and interval genetic algorithm (iGA) wavelength. PLS: partial least squares.

| Parameters | N | Method | Wavelength | Pretreatment | lv | $R^2_{cal}$ | $R^2_{val}$ | SEC | SECV |
|---|---|---|---|---|---|---|---|---|---|
| FR, % | 140 | Full-PLS | 256 | raw | 8 | 0.71 | 0.63 | 0.04 | 0.04 |
|  | 140 | iSPA-PLS | 100 | D2 | 9 | 0.78 | 0.72 | 0.03 | 0.04 |
|  | 140 | iGA-PLS | 25 | D2 | 10 | 0.72 | 0.66 | 0.04 | 0.04 |
| VM, % | 140 | Full-PLS | 256 | D1 | 8 | 0.89 | 0.86 | 1.74 | 1.95 |
|  | 140 | iSPA-PLS | 100 | SNV | 8 | 0.90 | 0.88 | 1.67 | 1.85 |
|  | 140 | iGA-PLS | 100 | raw | 9 | 0.89 | 0.86 | 1.75 | 1.96 |
| FC, % | 140 | Full-PLS | 256 | SNV | 9 | 0.88 | 0.85 | 1.59 | 1.82 |
|  | 140 | iSPA-PLS | 100 | SNV | 8 | 0.85 | 0.81 | 1.78 | 2.01 |
|  | 140 | iGA-PLS | 50 | SNV | 10 | 0.83 | 0.77 | 1.91 | 2.23 |
| Ash, % | 140 | Full-PLS | 256 | SNV | 9 | 0.93 | 0.91 | 2.18 | 2.48 |
|  | 140 | iSPA-PLS | 100 | SNV | 7 | 0.92 | 0.91 | 2.36 | 2.62 |
|  | 140 | iGA-PLS | 50 | D2 | 9 | 0.90 | 0.87 | 2.69 | 3.02 |

[iSPA] interval successive projections algorithm; [iGA] genetic algorithm; [raw] raw spectra; [SNV] standard normal variate; [D1] first derivative; [D2] second derivative; [SEC] standard error of calibration; [SECV] standard error of cross-calibration.

Figure 6a–d shows the result of the PLS model developed from different spectra pretreatments and numbers of iSPA wavelengths, including 12, 25, 50, 100, and 150 wavelengths, for the prediction of FR, VM, FC, and A, respectively. The results showed that the optimal model for predicting FR could be developed using D2 spectra and the selected 100 wavelengths. Meanwhile, VM, FC, and A could be developed using SNV spectra coupled with 100 wavelengths because this provided a low standard error of cross-calibration (SECV).

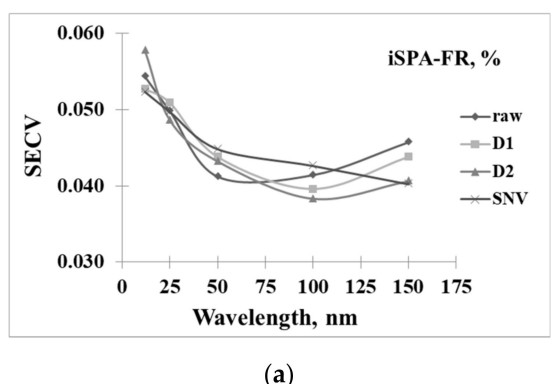

**(a)**

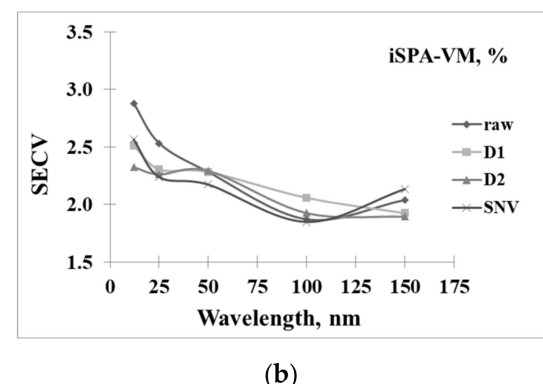

**(b)**

**Figure 6.** *Cont.*

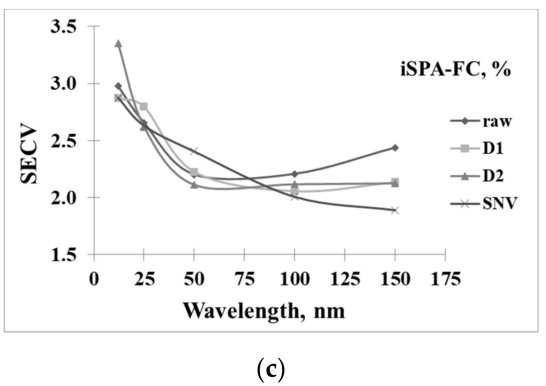

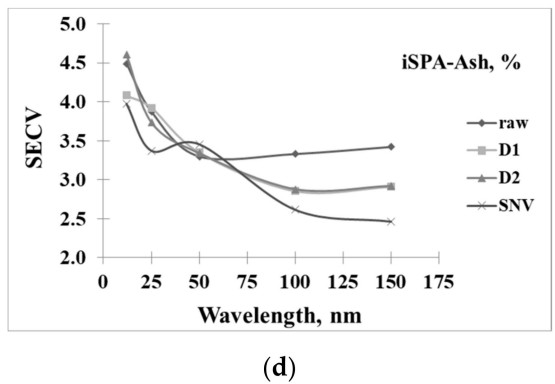

**Figure 6.** Wavelength selected by iSPA for 12, 25, 50, 100, and 150 wavelengths in prediction of (**a**) fuel ratio (FR, %), (**b**) volatile matter (VM, %), (**c**) fixed carbon (FC, %), and (**d**) ash content (A, %).

Figure 7a–d shows the results of the model generated using different spectra pretreatment methods and wavelengths selected from the iGA wavelengths, including 12, 25, 50, 100, and 150 wavelengths. The most effective model for predicting FR was developed using D2 spectra coupled with 25 wavelengths, where the SECV was 0.043% (Figure 7a). The optimal model for predicting VM was developed using raw spectra coupled with 100 wavelengths, where the SECV was 1.96% (Figure 7b). The optimal model for predicting FC was developed using SNV spectra coupled with 50 wavelengths, where the SECV was 2.23% (Figure 7c). The optimal model for predicting A was developed using D2 spectra coupled with 50 wavelengths, where the SECV was 3.02% (Figure 7d).

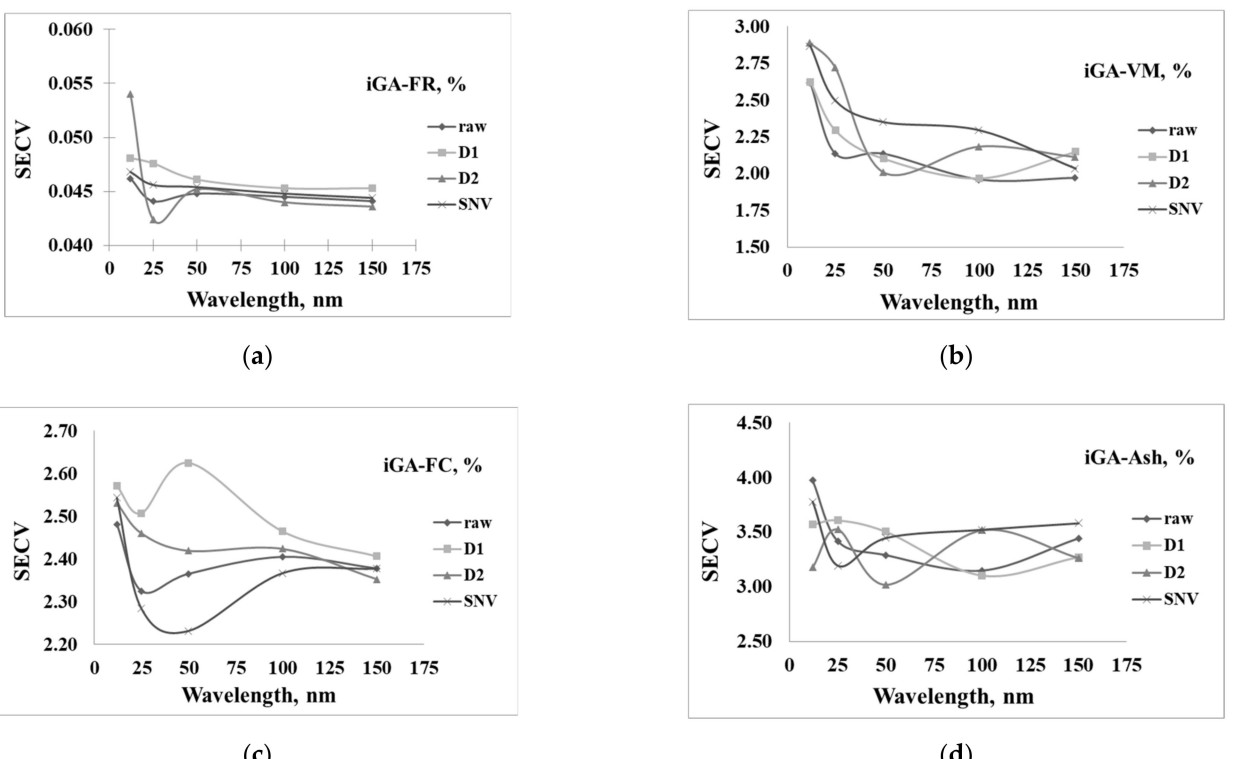

**Figure 7.** Wavelengths selected by iGA for 12, 25, 50, 100, and 150 wavelengths in prediction of (**a**) fuel ratio (FR, %), (**b**) volatile matter (VM, %), (**c**) fixed carbon (FC, %), and (**d**) ash content (A, %).

When comparing the full wavelengths, iSPA wavelengths, and iGA wavelengths, the iSPA was the best wavelength selection method because it provided the lowest SECV. Therefore, the most effective model could be developed with iSPA, 100 wavelengths, D2 spectra, and nine latent variables for the FR model; iSPA, 100 wavelengths, SNV spectra, and eight latent variables for the VM model; iSPA, 100 wavelengths, SNV spectra, and eight latent variables for the FC model; and iSPA-100 wavelengths, SNV spectra, and seven latent variables for the A model.

To confirm the accuracy, the PLS model was re-calculated and validated with the test set method, as the best result shows in Table 4 (selecting the best model from Table 3). The model of FR provided $R^2$, SEC, and SEP values of 0.75, 0.04, and 0.03, respectively. The VM model gave $R^2$, SEC, and SEP values of 0.82, 1.62, and 2.10%, respectively. The FC model had $R^2$, SEC, and SEP values of 0.81, 1.82, and 1.80%, respectively. Meanwhile, the A model provided $R^2$, SEC, and SEP values of 0.88, 2.27, and 2.53%, respectively. The SEP values were very similar to the SEC values, which means that these models had no over-fitting and that the NIR models were effective. In the author's opinion, with a low SEP of approximately 7.8% of the range of reference value (0.03 of 0.38) for FR, 7.6% of the mean value (2.10% of 27.57%) for VM, 8.0% of the mean value (1.8% of 22.40%) for FC, and 6.2% of the mean value (2.53% of 40.17%) for A, all of which were less than 10%, the model was acceptable for the approximation of the FR, VM, FC, and A of biomass pellets.

**Table 4.** The result of the best PLS regression model using iSPA wavelengths for the prediction of various biomass pellets in the calibration set and the validation set.

| Parameters | Calibration Set | | | | Validation Set | | | |
|---|---|---|---|---|---|---|---|---|
| | PLS Factor | N | $R^2$ | SEC | n | $r^2$ | SEP | RPD | Bias |
| FR, % | 9 | 106 | 0.76 | 0.04 | 34 | 0.75 | 0.03 | 1.97 | 0.01 |
| VM, % | 8 | 106 | 0.91 | 1.62 | 34 | 0.82 | 2.10 | 2.46 | 0.10 |
| FC, % | 8 | 106 | 0.86 | 1.82 | 34 | 0.81 | 1.80 | 2.32 | −0.39 |
| Ash, % | 7 | 106 | 0.93 | 2.27 | 34 | 0.88 | 2.53 | 3.17 | −0.44 |

R2: coefficient of determination of calibration; r2: coefficient of determination of validation; N: number of samples; PLS: partial least squares; SEC: standard error of calibration; SEP: standard error of prediction; RPD: ratio of prediction to deviation.

The scatter plots of the measured and predicted values of the validation set are shown in Figure 8a–d for FR, VM, FC, and A, respectively.

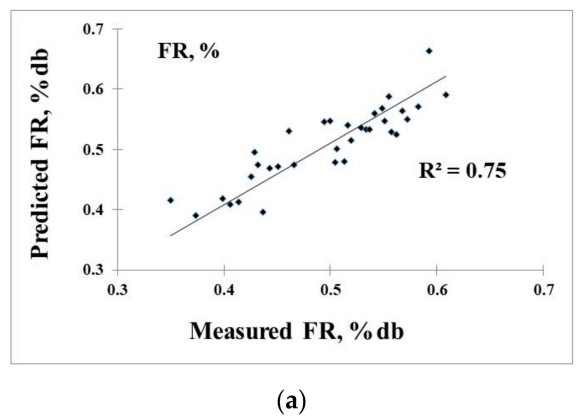

(**a**)

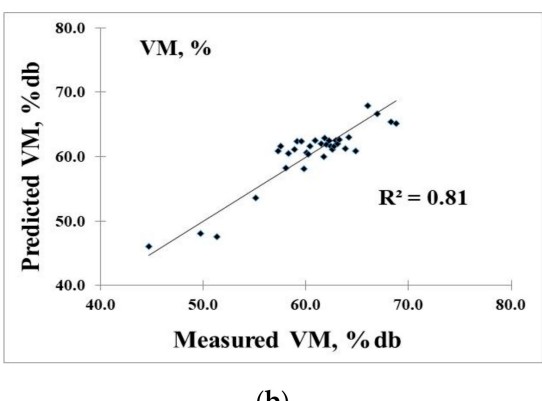

(**b**)

**Figure 8.** *Cont.*

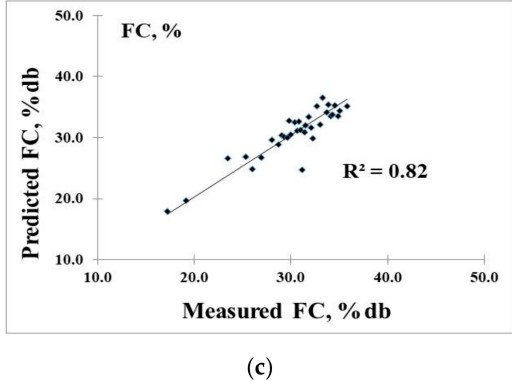

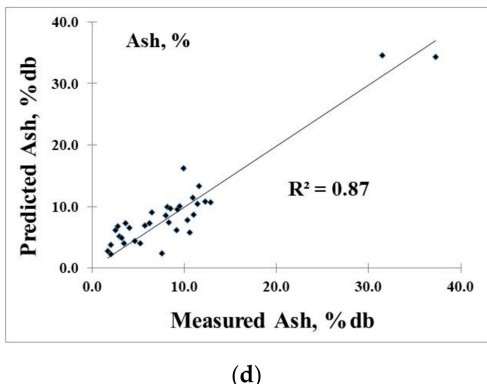

(**c**)    (**d**)

**Figure 8.** Scatter plots of measured value versus predicted value of 34 biomass pellets in the prediction of (**a**) fuel ratio (FR, %db), (**b**) volatile matter (VM, %db), (**c**) fixed carbon (FC, %db), and (**d**) ash content (A, %db).

After comparison to other works, the result was seen to be similar to that of Feng et al. [16], who reported that the SPA-PLSR model predicted the VM, FC, and ash with $R^2_{cal}$ values of 0.90, 0.85, and 0.94, respectively. Our accuracy showed a similar result to that of Sirisomboon et al. [43], who predicted the VM, FC, and A content of bamboo chips using Fourier Transform - Near Infrared (FT-NIR) spectroscopy, using a model that provided $R^2$ values of 0.81, 0.81, and 0.86, respectively.

### 3.4. Result Visualisation of FR, VM, FC, and A in the Distribution Map

Distribution maps representing a linear color scale that defines the differences in the FR, VM, FC, and A levels are shown in Figure 9. Figure 9a shows raw relative reflectance images of some biomass pellet types before prediction. The linear color scale in each pixel was predicted and then converted into color. The predictive distribution maps converted into 2D NIR images of FR, VM, FC, and A are shown in Figure 9b–e, respectively. The highest predicted values are shown by red, the middle values are shown by green, and the lowest values are shown by blue. Blue indicates that the predicted value was zero, which was found in the background. Other colors in the linear color scale varied according to the level of the predicted value. This visualization for the distribution of pellet properties could be suitable and intuitive for online industry application.

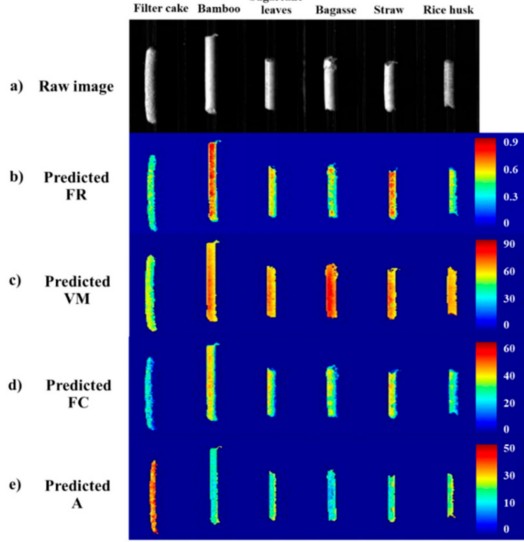

**Figure 9.** Predictive distribution maps of pellet quality of different biomass pellet types predicted by the iSPA-PLS model; (**a**) raw image, (**b**) predicted FR, (**c**) predicted VM, (**d**) predicted FC, and (**e**) predicted A.

## 4. Conclusions

The in-line prediction of the FR and proximate data of commercial biomass pellets using NIR-hyperspectral imaging was investigated. Standardized samples of commercial biomass pellets were collected for the experiment because of their unified form, comparable density, and mechanical durability, meaning that these factors did not affect the results of the model. The different spatial wavelengths, including full wavelength, iSPA and iGA wavelengths, and different spectra pretreatments, were studied. The optimal model for all parameters could be developed using iSPA-100 spatial wavelengths with D2 for FR and SNV spectra for VM, FC, and A. In practical applications, we could reduce the number of wavelengths from 256 to 100, which would reduce the cost of the line scanning spectrometer. The result demonstrated that good accuracy could be achieved in the prediction of FR, VM, FC, and A, suitable for application for quality screening. Therefore, the prediction of biomass pellets' property distribution maps could enable the control of the thermal conversion process and be intuitive for online industry applications. For example, a distribution map could be easily visualized and utilized at the stages of the drying, storage, and maybe mixing of raw materials. However, in order to further increase the model's accuracy, it should be developed with several types of biomass, e.g., woody, herbaceous, and fruit biomass. The model can be improved in the future to be used for classification in the standard EN ISO 17225-1, according to which the range of measured parameters in these groups can be specified to create this classification.

**Author Contributions:** Conceptualization, L.P. and J.P.; methodology, L.P. and J.P.; software, L.P.; validation, S.W., K.L. and P.S.; formal analysis, L.P.; investigation, J.P.; resources, J.P.; data curation, L.P.; writing—original draft preparation, L.P.; writing—review and editing, J.P. and P.S.; visualization, J.P.; supervision, J.P.; project administration, J.P.; funding acquisition, J.P. All authors have read and agreed to the published version of the manuscript.

**Funding:** This research was funded by Research and Academic Services, Khon Kaen University, Thailand, Research and Graduate Studies Khon Kaen University, Thailand and Research Grant for New Scholar, grant number MRG6280078 of the Thailand Research Fund (TRF), Thailand and the APC was funded by authors.

**Institutional Review Board Statement:** Not applicable.

**Informed Consent Statement:** Not applicable.

**Data Availability Statement:** Not report any data.

**Acknowledgments:** This work was supported by Research and Academic Services, Khon Kaen University, Thailand, Research and Graduate Studies Khon Kaen University, Thailand and Research Grant for New Scholar (MRG6280078) of the Thailand Research Fund (TRF), Thailand.

**Conflicts of Interest:** The authors declare no conflict of interest.

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
