# Peer review of "Machine Learning-Based Prediction of Selected Parameters of Commercial Biomass Pellets Using Line Scan Near Infrared-Hyperspectral Image"

_processes, doi:10.3390/pr9020316_

Round 1

Reviewer 1 Report

The article “Machine learning based prediction of fuel ratio and proximate data of commercial biomass pellets using line scan near infra-red-hyperspectral image” is focused on the use of near infra-red-hyperspectral imaging to predict the quality of pellet by means of machine learning based prediction models. The topic is in line with Processes aim and scope and the use of hyperspectral imaging coupled with machine learning technique is new for the pellet quality assessment. Unfortunately, English must be checked carefully and corrected throughout the manuscript, there are several errors and some sentences are not clear. This makes it difficult to assess the manuscript. The manuscript could be resubmitted and evaluated after this revision.

I add also some technical suggestions below.

Introduction:

English must be improved: e.g.: “Recently, the world requirement of renewable energy increase [1].” It should be “increased”

“In 2020, the use of renewable energy increases to 3.9% from the previous year [2].” Should be “increased” Biomass is considered as renewable energy because of abundant value and clean [3]. Not clear, please improve the sentence. And so on throughout the manuscript.

As “international standards” the relevant ISO Standards should be mentioned because regulating the pellet market at international level.

Based on my experience, biomass and biofuel come from various and different origins. So, the statement “Biomass are obtained from agricultural residues and fast-growing wood” used by the authors seems to be reductive. More information are needed on the samples: origin, management, etc. These biomass materials are variable and this aspect is important. Also the pellet quality would be important (durability, for example) to assess if the pelletisation process was correct for all the materials.

Authors stated about the disadvantages of traditional measurement methods, but rapidity of NIRS was only underlined. Even if rapidity is one of the most important and useful advantages of NIRS, I think it could be relevant to highlight other benefits of NIRS, to create a valid comparation between methods.

Results:

I understand that R2=coefficient of determination; r2= coefficient of determination of prediction set; R2cal and R2val=respectively, coefficient of determination in calibration and validation. So, what R2c refers to?

Section 3.4: Figure 8 shows the distribution map of pellet quality, but there is no explanation about the interpretation (not even in materials and methods). To better understanding, authors could add a short results explanation.

Conclusions:

“The result demonstrated that a good accuracy was found in prediction of FR, VM, FC and A and could be applied for quality assurance." Generally, with RPD over 3 you can use a prediction model for quality assurance. Your results show RPD in the range 2-3. Maybe it is better to write for quality screening.

Author Response

Manuscript ID: processes-1074502
Type of manuscript: Article
Title: Machine learning based prediction of fuel ratio and proximate data of commercial biomass pellets using line scan near infrared-hyperspectral image
Authors: Lakkana Pitak, Kittipong Laloon, Seree Wongpichet, Panmanas Sirisomboon, Jetsada Posom *

Dear Editor and reviewers

We thank you for your careful reading of the manuscript and helpful comments and suggestions. We have made revisions according to reviewer comment.

Our detailed, point-by-point responses to the editorial and reviewer comments are given below, whereas the corresponding revisions are marked in colored text in the revised manuscript file. Specifically, red text indicates changes made in response to the suggestions of reviewer.

We would like to thank you once again for your consideration of our work and inviting us to submit the revised manuscript. We look forward to hearing from you.

Best regards,

Jetsada Posom

Reviewer 1

The article “Machine learning based prediction of fuel ratio and proximate data of commercial biomass pellets using line scan near infra-red-hyperspectral image” is focused on the use of near infra-red-hyperspectral imaging to predict the quality of pellet by means of machine learning based prediction models. The topic is in line with Processes aim and scope and the use of hyperspectral imaging coupled with machine learning technique is new for the pellet quality assessment. Unfortunately, English must be checked carefully and corrected throughout the manuscript, there are several errors and some sentences are not clear. This makes it difficult to assess the manuscript. The manuscript could be resubmitted and evaluated after this revision.

I add also some technical suggestions below.

  • Thank you very much for your comment. The manuscript was edited by English grammar by English Lecturer.

Introduction:

English must be improved: e.g.: “Recently, the world requirement of renewable energy increase [1].” It should be “increased”

  • We corrected it as your suggestion, and change the word “increase” to “increased”.

 “In 2020, the use of renewable energy increases to 3.9% from the previous year [2].” Should be “increased” Biomass is considered as renewable energy because of abundant value and clean [3]. Not clear, please improve the sentence. And so on throughout the manuscript.

  • Thank you very much for your comment. We revised as throughout the manuscript (please see red color text in Introduction part).

As “international standards” the relevant ISO Standards should be mentioned because regulating the pellet market at international level.

  • Thank you for your comment. We add more information as “ For commercial purposes, the quality of biomass pellet must follow the international standards i.e., European standards for pellet quality [7]. The important parameters of biomass pellets in production process is density, strength and moisture content (MC) [9].”
  • This is indicated in Introduction (Page2 Line 11-12).

Based on my experience, biomass and biofuel come from various and different origins. So, the statement “Biomass are obtained from agricultural residues and fast-growing wood” used by the authors seems to be reductive. More information are needed on the samples: origin, management, etc. These biomass materials are variable and this aspect is important. Also the pellet quality would be important (durability, for example) to assess if the pelletisation process was correct for all the materials.

  • As your suggestion and we add more information as follows “Therefore, biomass feedstock can be produced and reused again and again. Biomass is obtained from dedicated energy crops, agricultural crop residues, forestry residues, wood processing residues, municipal waste, and wet waste [5]. This leads to biomass materials are different in physical properties, energy content and chemical content. Most of the biomass energy is used as combustion fuel. It is combusted to generate electricity which is a source of thermal energy for an industrial purpose [6]. Therefore, the quality of biomass is required for quality control of the thermal conversion process, commercial and pellet production as well.”
  • This is now indicated in the revised manuscript at Introduction part (Page1 Line 7-14).

Reference

[5] Biomass Resources. https://www.energy.gov/eere/bioenergy/biomass-resources. Access on 10 Jan, 2021.

Authors stated about the disadvantages of traditional measurement methods, but rapidity of NIRS was only underlined. Even if rapidity is one of the most important and useful advantages of NIRS, I think it could be relevant to highlight other benefits of NIRS, to create a valid comparation between methods.

  • Thank you very much again for your review. we add more information as follows “Nowadays, NIR spectroscopy is used in the quality measurement of biomass, such as MC and calorific value (CV) of biomass pellet of rice stalks, rice husks, mahogany wood, tea trees, tropical wood, pine wood, rubber wood, and mixtures of wood material [12], pyrolysis characteristics of milled bamboo [13], proximate data and lignocellulose components (cellulose, semi-cellulose, and lignin) of corn stover using NIRS [14], MC, CV, A and carbon content (C) of Miscanthus and Short Rotation Coppice Willow (SRCW) biomass [15], biomass pellet (wood, Miscanthus and herbaceous energy grasses) quality indices (moisture, carbon and ash contents and gross calorific value) [16]. It is noted that NIR spectroscopy can be applied for assessing the composition of analytes. It is an indirect method of measurement, and its accuracy cannot be equal to that of the laboratory method [Pitak et al., 2021 and Shaw et al., 1996]. The application NIR is acceptable if the error is not exceeded the control limit.”
  • This is placed in Introduction (Page2 Line 44-47).
  • Reference
  • Shaw, RA.; Kotowich, S.; Mantsch, HH.; Leroux, M. Quantitation of protein, creatinine, and urea in urine by near-infrared spectroscopy. Clin Biochem. 1996, 29(1), 11-9.
  • Pitak, L.; Sirisomboon, P.; Saengprachatanarug, K.; Wongpichet, S.; Posom, J. Rapid elemental composition measurement of commercial pellets using line-scan hyperspectral imaging analysis. Energy 2021, 220, 119698, 1-11.

Results:

I understand that R2=coefficient of determination; r2= coefficient of determination of prediction set; R2cal and R2val=respectively, coefficient of determination in calibration and validation. So, what R2c refers to?

  • We change the word “R2c” to “R2cal”.

Section 3.4: Figure 8 shows the distribution map of pellet quality, but there is no explanation about the interpretation (not even in materials and methods). To better understanding, authors could add a short results explanation.

  • Thank you very much for your comment. We add more explanation as follows

“3.4. Result visualization of FR, VM, FC and A in the distribution map

The distribution maps represented the linear color scale which defined the difference in FR, VM, FC and A level was demonstrated in Figure 8. Figure 8a shows raw relative reflectance image of some biomass pellet types before prediction. The linear color scale in each pixel was predicted and then transferred to color. The predictive distribution maps into 2D NIR images of FR, VM, FC and A were shown in Figure 8b, c, d, and e, respectively. The highest values for predicted value were shown by red, middle by green and the lowest by blue. Blue indicated that the predicted value was zero, which found in the background. Other colors in the linear color scale varied according to the level of predicted value. The visualization for the distribution of pellet properties could be suitable and intuitive for online industry application.”

  • This is placed in item 4. Result visualization of FR, VM, FC and A in the distribution map.

Conclusions:

“The result demonstrated that a good accuracy was found in prediction of FR, VM, FC and A and could be applied for quality assurance." Generally, with RPD over 3 you can use a prediction model for quality assurance. Your results show RPD in the range 2-3. Maybe it is better to write for quality screening.

  • Thank you very much for We change the word “quality assurance” to “quality screening”.

**Please see revised manuscript in the attachment.

Reviewer 2 Report

The manuscript proposed by the authors is interesting and valuable in terms of quick measurement of quality parameters of the raw material (VM, FC, A). The presented method replaces several other time-consuming methods. Such an evaluation on the production line allows for quick reactions of the staff, e.g. to change process parameters. However, the presented method allows to determine the parameters dependent on the raw material not on the agglomeration process. What is a disadvantage of the presented method. These parameters do not change during pelletization (maybe VM to a small extent if the pelletization is accompanied by high temperature) so once characterized, the raw material does not need to control these parameters on the line. An interesting aspect would be the application of this method to parameters dependent on the agglomeration process, i.e. the specific density (DE) and mechanical durability (DU) of the pellets - such information obtained quickly on an ongoing basis during production would allow for a quick reaction to improve these parameters. Nevertheless, the article presents an interesting application of the NIR method to determine the parameters of the raw material, which can be used as a quick evaluation of the raw material, e.g. at the stage of purchasing the raw material (quick evaluation of the delivered batch of raw material, identification of the type of raw material), selection of components of the mixture for production or its evaluation or at the stage of drying which parameters, mainly the applied temperature, can change the raw material parameters. The presented method is interesting, however, its application proposed by the authors is not correct. The subject is in line with a Journal scope. It can be published after major corrections concerning mainly the "Introduction" and "Material and Methods" chapters.

1 Introduction

Page1 Line 3-5 It is true, however, all operations concerning the acquisition, processing and transport of biomass are carried out on the basis of conventional energy sources accompanied by CO2 emissions.

Page1 Line 12-13 It is not true that pellets have a higher energy content in comparison to the raw material from which they are made. The pelletisation process does not change the energy parameters of the raw material, nor its calorific value, nor the parameters that the authors have studied, i.e. VM, A, FC. This should be absolutely stated here.

Page2 First paragraph - There should be a clear separation between the quality parameters depending on the raw material (precisely those which the authors measured) which can be marked on the raw material before pelletisation, and the parameters depending on the pelletisation process (DU, DE) which the authors did not mention at all. The authors identify the quality parameters of the raw material with the quality parameters of pellets, completely ignoring the quality parameters dependent on the pelletization process (DU,DE). They determine the quality of pellets. I propose to review the pellet quality standards.

Page2 second and third paragraph - once again I emphasize these parameters (VM, FC, A) do not change during the pelletization and their measurement during the production is unreasonable and does not bring any information to improve the quality parameters dependent on this process (DU, DE). Such measurement would be important at the stage of drying the raw material or mixing different raw materials.  At the pelletising stage, the measurement of density, strength and humidity would be important information. It should be remembered that the moisture content of the pellet immediately after pelletization is different from that of the final pellet (stabilization and cooling of the pellet causes a decrease in moisture). Quick measurement of these three parameters would give valuable information to control the production process

Page 3 line 6-8 these research, or rather the measurement of these parameters will be useful, but not at the pelletisation stage but at the earlier stages, especially at the stage of drying or storing the raw material

  1. Material and Methods

Page 4 line 7-8 Why the moisture content was not measured, without this information it is not possible to determine the other parameters of the raw material in the dry state and thus it is impossible to compare the raw materials. Moisture content is the basic parameter that should be determined during the characterization of biomass because other parameters strictly depend on it.  Especially that the authors state that the NIR also allows for this [reference 18]

Also, the pellet density was not determined and this may affect the result of other measurements. Going further, will a measurement made on pellet and raw material in the form of dust give the same results?

  1. Results and Discision

3.2 Reference Value – why the Authors did not give the results of measured parameters for individual samples - perhaps the samples could be grouped according to these values

Page 11 Fig. 8 – on the Raw Image the surface of the sample is irregularly illuminated - does this not affect the accuracy of the results?

  1. Conclusion

Line 8-10 This method allows to characterize the raw material in terms of the content of the tested parameters, but it cannot be used to assess the quality of pellets on the production line because these parameters do not depend on the pelletization process.It can be used at the stage of drying, storage, maybe mixing of raw materials.

Author Response

Manuscript ID: processes-1074502
Type of manuscript: Article
Title: Machine learning based prediction of fuel ratio and proximate data of commercial biomass pellets using line scan near infrared-hyperspectral image
Authors: Lakkana Pitak, Kittipong Laloon, Seree Wongpichet, Panmanas Sirisomboon, Jetsada Posom *

Dear Editor and reviewers

We thank you for your careful reading of the manuscript and helpful comments and suggestions. We have made revisions according to reviewer comment.

Our detailed, point-by-point responses to the editorial and reviewer comments are given below, whereas the corresponding revisions are marked in colored text in the revised manuscript file. Specifically, red text indicates changes made in response to the suggestions of reviewer.

We would like to thank you once again for your consideration of our work and inviting us to submit the revised manuscript. We look forward to hearing from you.

Best regards,

Jetsada Posom

Reviewer 2

The manuscript proposed by the authors is interesting and valuable in terms of quick measurement of quality parameters of the raw material (VM, FC, A). The presented method replaces several other time-consuming methods. Such an evaluation on the production line allows for quick reactions of the staff, e.g. to change process parameters. However, the presented method allows to determine the parameters dependent on the raw material not on the agglomeration process. What is a disadvantage of the presented method. These parameters do not change during pelletization (maybe VM to a small extent if the pelletization is accompanied by high temperature) so once characterized, the raw material does not need to control these parameters on the line. An interesting aspect would be the application of this method to parameters dependent on the agglomeration process, i.e. the specific density (DE) and mechanical durability (DU) of the pellets - such information obtained quickly on an ongoing basis during production would allow for a quick reaction to improve these parameters. Nevertheless, the article presents an interesting application of the NIR method to determine the parameters of the raw material, which can be used as a quick evaluation of the raw material, e.g. at the stage of purchasing the raw material (quick evaluation of the delivered batch of raw material, identification of the type of raw material), selection of components of the mixture for production or its evaluation or at the stage of drying which parameters, mainly the applied temperature, can change the raw material parameters. The presented method is interesting, however, its application proposed by the authors is not correct. The subject is in line with a Journal scope. It can be published after major corrections concerning mainly the "Introduction" and "Material and Methods" chapters.

  • Thank you very much. We revised as your comment.

1 Introduction

Page1 Line 3-5 It is true, however, all operations concerning the acquisition, processing and transport of biomass are carried out on the basis of conventional energy sources accompanied by CO2 emissions.

  • We added more explanation, this is indicated in Introduction

Page1 Line 12-13 It is not true that pellets have a higher energy content in comparison to the raw material from which they are made. The pelletisation process does not change the energy parameters of the raw material, nor its calorific value, nor the parameters that the authors have studied, i.e. VM, A, FC. This should be absolutely stated here.

  • We edited as follows “Presently, a biomass pattern used for combustion includes ground biomass, biomass chip and pellet. Ground and wood chip are easy to prepare; however, they are difficult to store and transport and the variation of moisture content (MC) is high. While biomass pellets provide many advantages such as having a consistent size, lower moisture content, higher in calorific density, and convenient for transportation [6].”
  • This is indicated in Introduction (Page2 Line 1-5)

Page2 First paragraph - There should be a clear separation between the quality parameters depending on the raw material (precisely those which the authors measured) which can be marked on the raw material before pelletisation, and the parameters depending on the pelletisation process (DU, DE) which the authors did not mention at all. The authors identify the quality parameters of the raw material with the quality parameters of pellets, completely ignoring the quality parameters dependent on the pelletization process (DU,DE). They determine the quality of pellets. I propose to review the pellet quality standards.

  • Thank you very much for your comment. We then added more information to Introduction as follows “To confirm the stable development of the biomass pellet, the quality of commercial pellets must be maintained quality in the production line, which depended on raw materials. Therefore, rapid and accurate method in determination of pellet quality could be used to select the pellet products that providing a good combustion performance. This can achieve in the efficient utilization of biomass.”
  • This is explained in Page2 Line 5-10.

Page2 second and third paragraph - once again I emphasize these parameters (VM, FC, A) do not change during the pelletization and their measurement during the production is unreasonable and does not bring any information to improve the quality parameters dependent on this process (DU, DE). Such measurement would be important at the stage of drying the raw material or mixing different raw materials.  At the pelletising stage, the measurement of density, strength and humidity would be important information. It should be remembered that the moisture content of the pellet immediately after pelletization is different from that of the final pellet (stabilization and cooling of the pellet causes a decrease in moisture). Quick measurement of these three parameters would give valuable information to control the production process

  • We edited as follows: “For commercial purposes, the quality of biomass pellet must follow the international standards i.e., European standards for pellet quality [8]. The important parameters of biomass pellets in production process is density, strength and moisture content (MC) [9].
  • For utilization of biomass, the parameters of proximate analysis and calorific value is essential [10]. Biomass pellets with a high volatile matter (VM) can be flammable and could be prepared for pyrolysis and gasification, meanwhile biomass pellets with high fixed carbon (FC) give high combustion heating [7] and could be used for direct combustion and slow pyrolysis. The ratio of FC to VM is called “fuel ratio (FR)”, and is an important parameter used to manage the ratio of syngas and combustion air [11]. From the study of the burning rate of coal, it was found that the coal can be burned easily if the fuel ratio is lower than 1.5 and it is difficult to ignite if the FR is higher than 1.5. Therefore, the knowledge of the FR can improve the efficiency of thermal energy combustion [12]. Ash (A) is residue after combustion, it is a factor used for designing combustion furnaces [10]. Low A content in biomass material can reduce the cost of disposal [7].”
  • It is indicted in Page2 Line 10-24.

Page 3 line 6-8 these research, or rather the measurement of these parameters will be useful, but not at the pelletisation stage but at the earlier stages, especially at the stage of drying or storing the raw material

  • We edited as your suggestion: “The main objective of this research was to predict FR and proximate data of commercial biomass pellets using NIR hyperspectral image. The sub objectives include: to compare the model performance developed from different spatial wavelength (among full wavelength, interval successive projections algorithm (iSPA) and internal genetic algorithm (iGA) wavelengths) and different spectral pretreatment method (among raw), first derivative (D1), second derivative (D2), and standard normal variate (SNV), and to select the optimal prediction models. This research will benefit biomass pellet trading and thermal conversion processes in order to achieve better efficiency and the pelletization process at the stage of drying or storing the raw material.”
  • This indicted in Page3 Line 22-24

  1. Material and Methods

Page 4 line 7-8 Why the moisture content was not measured, without this information it is not possible to determine the other parameters of the raw material in the dry state and thus it is impossible to compare the raw materials. Moisture content is the basic parameter that should be determined during the characterization of biomass because other parameters strictly depend on it.  Especially that the authors state that the NIR also allows for this [reference 18]

  • Thank you very much for your comment. This is our mistake. In the past, we thought that the MC is not necessary if the FR, VM, FC and A on the dry basis is known because in commercial standard the MC must be less 10%. As your comment, the moisture content (MC) must be measured because other parameters strictly depend on it. However, we cannot revise in time due to we have to set new experiment.
  • Then we add your suggestion in Conclusion, as follows: MC is the basic parameter that should be determined during the characterization of biomass because other parameters strictly depend on it. Therefore, the MC model of biomass pellet could be created in the future.

Also, the pellet density was not determined and this may affect the result of other measurements. Going further, will a measurement made on pellet and raw material in the form of dust give the same results?

  • Thank you very much for We have to study more about the pellet production process and we are going to study about the pellet density model in the future. Unfortunately, in ten days for revision, we cannot study in time. We have to apologize.
  1. Results and Discision

3.2 Reference Value – why the Authors did not give the results of measured parameters for individual samples - perhaps the samples could be grouped according to these values

  • Thank you very much for We added a Figure 5 to the manuscript to give more information.
  • And we then give more explanation as “ Table 1 shows the statistical properties of all biomass pellets including FR, VM, FC and A. FR, VM, FC, and A were ranged301 to 0.677; 39.2 to 66.77 %; 12.83 to 35.23%; and -0.09 to 41.57%, respectively. Figure 5 shows the average FR, VM, FC and A of the biomass pellets. The proximate data of biomass pellet varies with different varieties. The results were similar to what was reported by Feng et al. [16]. VM was the major component of the dried biomass samples. The VM was ranged from 68.72% to 89.04%, while A was ranged from 0.26 to 15.94% and FC varied from 10.39% to 21.22%. ”
  • This is indicated in page7 and 8. 
  • (Please see revised manuscript in the attachment).

Page 11 Fig. 8 – on the Raw Image the surface of the sample is irregularly illuminated - does this not affect the accuracy of the results?

  • Thank you for your comment. On the Raw Image the surface of the sample is irregularly illuminated, means that there was the scattering problem resulting from the differences in the shiny samples and sample surfaces. Therefore, spectral pre-processing step was required to solve the scattering problem. For example, the first derivative (D1) and second derivative (D2) were able to eliminate the baseline offset problem. The SNV can decrease the multiplicative scattering effects [Romìa and BernaÌ€rdez, 2009; Posom et al., 20]. The spectra pre-treatment helps to increase the model accuracy.
  • It is placed in Page5 Line 16-19
  • Reference

Romìa, M. B.; BernaÌ€rdez, M. A. Infrared spectroscopy for food quality analysis and control; Academic Press 2009, pp. 74−75.

J Posom, J Klaprachan, K Rattanasopa, P Sirisomboon, K Saengprachatanarug, and S Wongpichet. Predicting Marian Plum Fruit Quality without Environmental Condition Impact by Handheld Visible−Near-Infrared Spectroscopy. ACS Omega 2020, 5, 43, 27909–27921.

  1. Conclusion

Line 8-10 This method allows to characterize the raw material in terms of the content of the tested parameters, but it cannot be used to assess the quality of pellets on the production line because these parameters do not depend on the pelletization process. It can be used at the stage of drying, storage, maybe mixing of raw materials.

  • Thank you very much again for your suggestion. We revised as follows “In-line prediction of FR and proximate data of commercial biomass pellets using NIR- hyperspectral image were investigated. The different spatial wavelength, including full wavelength, iSPA and iGA wavelength and different spectra pretreatment were studied. The optimal model for all parameters could be developed using 100-iSPA spatial wavelengths with D2 for FR, and SNV spectra for VM, FC and A. In application, we can reduce the number of wavelengths from 256 to 100, which reduces the cost of line scanning spectrometer. The result demonstrated that a good accuracy was found in prediction of FR, VM, FC and A and could be applied for quality screening. Therefore, the prediction of biomass pellets property distribution map could be able to control the thermal conversion process and intuitive for online industry application. For example, distribution map can be easily visualized and can be utilized at the stage of drying, storage, maybe mixing of raw materials. These results could be a guide for future applications, such as for trading and operational control of energy conversion systems.”

*Please see revised manuscript in the attachment .

Round 2

Reviewer 2 Report

Authors significantly improved the manuscript taking into account almost all my comments and suggestions. The article can be published, however it still needs some corrections.

I suggest you change the title: „Machine learning based prediction of selected parameters of commercial biomass pellets using line scan near infra-red-hyperspectral image”. It will be shorter, the information what parameters are taken into account is in the article text and keywords

  1. Material and Methods

Some important aspects should be more emphasized at the beginning of this chapter e.g.:

The investigated parameters (FR, VM, FC, A) are determined at the stage of assessing the suitability of biomass for energy purposes and for conversion into biofuels [it can be found in:

Anita Ierna Giovanni Mauromicaleb 2010 Cynara cardunculus L. genotypes as a crop for energy purposes in a Mediterranean environment https://doi.org/10.1016/j.biombioe.2010.01.018

Karbowniczak A., Hamerska J., Wróbel M., Jewiarz M., NÄ™cka K. (2018) Evaluation of Selected Species of Woody Plants in Terms of Suitability for Energy Production. In: Mudryk K., Werle S. (eds) Renewable Energy Sources: Engineering, Technology, Innovation. Springer Proceedings in Energy. Springer, Cham pp 735-742 doi.org/10.1007/978-3-319-72371-6_72

Havrland B., Ivanova T., Lapczynska-Kordon B., Kolarikova M. (2013) Comparative Analysis of Bio-Raw Materials and Biofuels. In Proceedings of 12th International Scientific Conference Engineering for Rural Development, Jelgava, pp. 541-544 Available at: http://www.tf.llu.lv/conference/proceedings2013/Papers/100_Havrland_B.pdf or other]

The model developed in the future will find an ideal application in such cases. However, in order to develop the first version of the model, it was decided to start with measurements on biomass in the form of pellets, which have standardized parameters such as specific density of about 1000 kg/m3, mechanical durability of about 95%. This uniformity will facilitate the measurement of selected parameters (FR, VM, FC, A).

The main weakness of the article regarding the lack of measurement of the moisture content of the test material should be explained, for example, as follows:

Taking pellet as a test material allowed to assume that its moisture content is in the range of 6-10%. Therefore, the model developed assumes MC as a constant that does not affect the result obtained. After verification, when the model works, it will be developed for application to a wider range of biomass forms (sawdust, chips, etc.). In this case, the moisture content of the raw material will be taken into account as a predicted parameter. The literature clearly indicates that the value of such parameters as VM, FC, A changes e.g. during drying, which means that it clearly depends on the moisture content. it can be found in:

[Lapczynska-Kordon B., Krzysztofik B., Sobol Z. (2018) Quality of dried cauliflower according to the methods and drying parameters. In BIO Web of Conferences Vol. 10,(02017). EDP Sciences. DOI: 10.1051/bioconf/20181002017, or other]

  1. Results and Discision

The table 1 and figure 5, in the title indicate that these are values related to the „as recieved” state of the material

The results for ash would the accuracy be so high if not for the high ash sample? Especially that the amount of ash generated during combustion depends on the origin of the biomass fuel and is mainly under 1% (wood biomass), under 3% but sometimes up to 10% (agro biomass). Confirmation can be found e.g.; in:

[Vassilev, S. V.; Baxter, D.; Vassileva, C.G. An Overview of the Behaviour of Biomass during Combustion: Part II. Ash Fusion and Ash Formation Mechanisms of Biomass Types. Fuel 2014]

  1. Conclusion

Here again, it should be clearly indicated that the pellet was chosen because of its unified form, comparable density and mechanical durability, therefore these factors do not affect the results of the model. Further development of the model for other forms of biomass (wood chips, dust, etc.) is needed, because it is only with them that the model can find practical application

The question is whether in further studies, in order to increase accuracy, you should try to develop dedicated models for given types of biomass, e.g. woody, herbaceous, fruit biomass, etc., according to the classification contained in the standard EN ISO 17225-1. Measured parameters in these groups have specific ranges which is one of the reasons (besides the origin of the biomass) for creating this classification.

Author Response

Manuscript ID: processes-1074502
Type of manuscript: Article
Title: Machine learning based prediction of fuel ratio and proximate data of commercial biomass pellets using line scan near infrared-hyperspectral image
Authors: Lakkana Pitak, Kittipong Laloon, Seree Wongpichet, Panmanas Sirisomboon, Jetsada Posom *

We thank you for your careful reading of the manuscript and helpful comments and suggestions. We have made revisions according to editor and reviewer comment. The red text indicates changes made in response to the suggestions of reviewer.

All authors have approved the manuscript and contributed significantly for the paper. We would like to thank you once again for your consideration of our work and inviting us to submit the revised manuscript. We look forward to hearing from you.

Best regards,

Jetsada Posom

Reviewer 2

Authors significantly improved the manuscript taking into account almost all my comments and suggestions. The article can be published, however it still needs some corrections.

I suggest you change the title: „Machine learning based prediction of selected parameters of commercial biomass pellets using line scan near infra-red-hyperspectral image”. It will be shorter, the information what parameters are taken into account is in the article text and keywords

  • Thank you very much for your suggestion, we revised as your comment.
  1. Material and Methods

Some important aspects should be more emphasized at the beginning of this chapter e.g.:

The investigated parameters (FR, VM, FC, A) are determined at the stage of assessing the suitability of biomass for energy purposes and for conversion into biofuels [it can be found in:

Anita Ierna Giovanni Mauromicaleb 2010 Cynara cardunculus L. genotypes as a crop for energy purposes in a Mediterranean environment https://doi.org/10.1016/j.biombioe.2010.01.018

Karbowniczak A., Hamerska J., Wróbel M., Jewiarz M., NÄ™cka K. (2018) Evaluation of Selected Species of Woody Plants in Terms of Suitability for Energy Production. In: Mudryk K., Werle S. (eds) Renewable Energy Sources: Engineering, Technology, Innovation. Springer Proceedings in Energy. Springer, Cham pp 735-742 doi.org/10.1007/978-3-319-72371-6_72

Havrland B., Ivanova T., Lapczynska-Kordon B., Kolarikova M. (2013) Comparative Analysis of Bio-Raw Materials and Biofuels. In Proceedings of 12th International Scientific Conference Engineering for Rural Development, Jelgava, pp. 541-544 Available at: http://www.tf.llu.lv/conference/proceedings2013/Papers/100_Havrland_B.pdf or other]

The model developed in the future will find an ideal application in such cases. However, in order to develop the first version of the model, it was decided to start with measurements on biomass in the form of pellets, which have standardized parameters such as specific density of about 1000 kg/m3, mechanical durability of about 95%. This uniformity will facilitate the measurement of selected parameters (FR, VM, FC, A).

The main weakness of the article regarding the lack of measurement of the moisture content of the test material should be explained, for example, as follows:

Taking pellet as a test material allowed to assume that its moisture content is in the range of 6-10%. Therefore, the model developed assumes MC as a constant that does not affect the result obtained. After verification, when the model works, it will be developed for application to a wider range of biomass forms (sawdust, chips, etc.). In this case, the moisture content of the raw material will be taken into account as a predicted parameter. The literature clearly indicates that the value of such parameters as VM, FC, A changes e.g. during drying, which means that it clearly depends on the moisture content. it can be found in:

[Lapczynska-Kordon B., Krzysztofik B., Sobol Z. (2018) Quality of dried cauliflower according to the methods and drying parameters. In BIO Web of Conferences Vol. 10,(02017). EDP Sciences. DOI: 10.1051/bioconf/20181002017, or other]

  • Thank you very much as again for your extremely suggestion. We then added more about sample information and background at the beginning of chapter “Materials and methods” as follows
  • 1. Investigated parameters

The main part of this study is based on estimation of the selected important parameter of biomass pellet in terms of suitability for energy production. Therefore, the investi-gated parameters (FR, VM, FC, and A) are determined at the stage of assessing the suitability of biomass for energy purposes and for conversion into biofuel. The above mentioned standardized biomass pellet samples which have standardized parameters such as specific density of about 1000 kg/m3, mechanical durability of about 95%, and moisture content <10% were taken to experiment which the FR, VM, FC, and A were determined. This uniformity facilitates the measurement of NIR hyperspectral imaging and selected parameters.

2.2. Sample

A total of 140 biomass pellet samples which were material obtained from wood pellet industry. The biomass pellets including filter cake (11 pellets), Leucaena leucocepphala (9 pellets), bamboo (9 pellets), cassava rhizome (15 pellets), bagasse (14 pellets), sugarcane leaves (15 pellets), straw (15 pellets), rice husk (14 pellets), euca-lyptus bark (11 pellets), Napier grass (13 pellets) and corn cob (14 pellets), were used for experiment. Taking pellet as a test material assumed that its moisture content is in the range of 6-10%. Therefore, the model developed assumed MC as a constant that does not affect the result obtained. They were pelletized using pelletizer machine (KN-D-200, Tianjin, China). The biomass pellet was kept in zipper bag and stored in room at temperature 25 ± 2 °C.

  1. Results and Discision

The table 1 and figure 5, in the title indicate that these are values related to the „as recieved” state of the material

The results for ash would the accuracy be so high if not for the high ash sample? Especially that the amount of ash generated during combustion depends on the origin of the biomass fuel and is mainly under 1% (wood biomass), under 3% but sometimes up to 10% (agro biomass). Confirmation can be found e.g.; in:

[Vassilev, S. V.; Baxter, D.; Vassileva, C.G. An Overview of the Behaviour of Biomass during Combustion: Part II. Ash Fusion and Ash Formation Mechanisms of Biomass Types. Fuel 2014]

  • From the Table 1 and Figure 5, we would like to inform that the VM, FR and A were calculated in dry basis. Then the VM, FC and A increase a bit. And we confirm that results for ash is correct due to the high ash sample.

Then we added “dry basis (db%)” to the title of Table 1 and Figure 5.

  1. Conclusion

Here again, it should be clearly indicated that the pellet was chosen because of its unified form, comparable density and mechanical durability, therefore these factors do not affect the results of the model. Further development of the model for other forms of biomass (wood chips, dust, etc.) is needed, because it is only with them that the model can find practical application

The question is whether in further studies, in order to increase accuracy, you should try to develop dedicated models for given types of biomass, e.g. woody, herbaceous, fruit biomass, etc., according to the classification contained in the standard EN ISO 17225-1. Measured parameters in these groups have specific ranges which is one of the reasons (besides the origin of the biomass) for creating this classification.

  • As your suggestion we improve the conclusion as

“4. Conclusions

In-line prediction of FR and proximate data of commercial biomass pellets using NIR- hyperspectral image were investigated. The standardized biomass pellet samples as commercial pellets were collected for experiment because of its unified form, com-parable density and mechanical durability, therefore these factors do not affect the results of the model. The different spatial wavelength, including full wavelength, iSPA and iGA wavelength and different spectra pretreatment were studied. The optimal model for all parameters could be developed using 100-iSPA spatial wavelengths with D2 for FR, and SNV spectra for VM, FC and A. In application, we can reduce the num-ber of wavelengths from 256 to 100, which reduces the cost of line scanning spectrom-eter. The result demonstrated that a good accuracy was found in prediction of FR, VM, FC and A and could be applied for quality screening. Therefore, the prediction of bio-mass pellets property distribution map could be able to control the thermal conversion process and intuitive for online industry application. For example, distribution map can be easily visualized and can be utilized at the stage of drying, storage, maybe mix-ing of raw materials. Moreover, in order to increase the model accuracy, the model should be developed with several types of biomass, e.g. woody, herbaceous, fruit bio-mass, etc. The model can be improved in the further to be used for classification in the standard EN ISO 17225-1, which the range of measured parameters in these groups can be specified for creating this classification.”
